# Metabolic Syndrome Is Associated with Poor Omicron Infection Prognosis While Inactivated Vaccine Improves the Outcome of Coronavirus Disease 2019 among Chinese Inhabitants: A Retrospective Observational Study from a Chinese Municipality

**DOI:** 10.3390/vaccines11101554

**Published:** 2023-09-30

**Authors:** Ying Liu, Dong Chen, Junfeng Li, Wei Wang, Rongfeng Han, Shanshan Cui, Suqing Bao

**Affiliations:** 1Endocrinology Department, Tianjin First Central Hospital, Nankai University, Tianjin 300192, China13702036711@163.com (W.W.); 15522831627@163.com (R.H.); cuishanshantj@163.com (S.C.);; 2Graduate School, Beijing Hospital of Traditional Chinese Medicine, Capital Medical University, Beijing 100010, China; chendong1319@foxmail.com

**Keywords:** COVID-19, Omicron, MetS, COVID-19 severity, inactivated vaccines

## Abstract

Coronavirus disease 2019 (COVID-19) and metabolic syndrome (MetS) are currently highly prevalent diseases worldwide. Studies on clinical outcomes of patients with Omicron and MetS, especially after vaccination with an inactivated vaccine are limited. Herein, we explored the relationship between MetS and the outcome of Omicron infection. Study Design: This was a retrospective observational study. Methods: This study recruited 316 individuals with Omicron infection. The inpatient data from between 8 January and 7 February 2022 were obtained from designated isolation hospitals in Tianjin, China. Hierarchical and multivariable analysis was conducted on age, gender, number of complications, and vaccination status. Results: Among the 316 study participants, 35.1% were diagnosed with MetS. The results showed that MetS was strongly associated with Intensive Unit Care (ICU) admission, Polymerase Chain Reaction (PCR) re-positivity, and severe COVID-19. The ICU admission rates of the unvaccinated individuals, those who received two-dose and full vaccination (3 doses), were 66.7%, 19.2%, and 0, respectively (*p* < 0.01). Two-dose and three-dose vaccinations significantly reduced PCR re-positivity. Conclusions: In summary, MetS increases the risk of ICU admission, PCR re-positivity, and severe COVID-19. MetS is a composite predictor of poor outcomes of Omicron infection. Two shots of inactivated vaccine, specifically three doses, effectively protect against Omicron even in the high-risk group.

## 1. Introduction

The Omicron variant, first detected in South Africa in November 2021, has spread rapidly across the globe. The Omicron SARS-CoV-2 variant has mutations in the receptor binding domain (RBD) of the spike (S) protein, increasing protein affinity with the human angiotensin-converting enzyme 2 (ACE2) that improves the transmissibility of the virus [1]. The Omicron variant with mutations in the S protein cannot be recognized and neutralized by the immune system [2]. The currently approved COVID-19 vaccines induce immune responses against other viral proteins other than spike proteins. The clinical protective efficacy of the vaccine, specifically among high-risk groups, is an important concern.

Metabolic syndrome (MetS) is a chronic metabolic condition highly common globally. In most countries, nearly 1/5 or more adults suffer MetS [3]. Recent research revealed that the prevalence of MetS in China is as high as 31.1% and has been increasing over the recent years [4]. MetS is clinically characterized by obesity, hypertension, hyperglycemia, and dyslipidemia. Increasing evidence has linked MetS to severe coronavirus disease 2019 (COVID-19) [5,6,7,8,9,10]. Specifically, MetS promotes chronic inflammation in the human body [11] by increasing the circulation of C-reactive protein (CRP), thrombosis-related and pro-inflammatory protein, and interleukin 6 [12]. MetS has been proposed as a predictor of poor outcomes of COVID-19 [13]. The Omicron variant is a highly transmissible but relatively less pathogenic SARS-CoV-2 variant. All vaccines are safe and effective tools that prevent severe COVID-19, hospitalization, and death against all variants of concern. However, the quality of evidence significantly varies. The relationship between MetS and the prognosis of infection by Omicron variant infection, specifically in patients vaccinated by the inactivated SARS-CoV-2 vaccine, is unclear.

Over the past 2 years, mass vaccination programs have been rolled out globally. In the Chinese mainland, the most administered vaccines are inactivated vaccines. As of 8 January 2022, when Omicron first emerged in Tianjin, up to 93.2% of its residents had been vaccinated to a varied extent [14]. Despite the generally high vaccination rates in the city, the variant spread quickly. We found inconsistent prognosis, particularly in patients with metabolic disorders. This study sought to investigate the relationship between MetS and clinical outcomes of patients with Omicron infection, focusing on ICU, admission, PCR re-positivity, and disease severity. We also investigated mechanisms by which age, gender, and other underlying patient complications influence, the relationship between MetS and clinical outcome of patients infected with Omicron. Furthermore, this work investigated the relationship between MetS features (e.g., hypertension, diabetes mellitus (DM), obesity, and hyperlipidemia) and the outcomes of Omicron infection. We explored the protective effect of the COVID-19 vaccine against Omicron infection in patients with MetS.

## 2. Methods

This retrospective and observational study analyzed clinical and demographic data on the 316 Omicron patients reported by Tianjin Municipal Health Commission between 8 January and 7 February 2022. The participants were 18 years or older, and the Omicron test was performed using nasopharyngeal swabs (Figure 1). These data were extracted from the inpatient medical records at Tianjin Haihe Hospital and Tianjin First Central Hospital, hospitals designated for isolating Omicron patients. This study was approved by the Ethics Committee of Tianjin’s First Central Hospital (Filing number: HHL2022005-EC-1).

All patients, including asymptomatic and mild cases, were hospitalized in Tianjin Haihe Hospital upon positive PCR results. Patients were discharged from Tianjin Haihe Hospital based on the following criteria: (1) restored body temperature and staying normal for over 3 days; (2) significantly relieved respiratory symptoms; (3) acute exudation substantially resolved on imaging study of the lungs; and (4) negative on two consecutive PCR tests (at an interval of at least 24 h) of samples collected from the respiratory tract. For patients whose PCR assays remained positive for over 4 weeks after criteria 1, 2, and 3 had been met, antibody assay and virus culture were applied to assess the risk of transmission before deciding whether these patients could be discharged.

Subsequently, the discharged patients from Tianjin Haihe Hospital were transferred to Tianjin First Central Hospital for at least 14 days under medical observation. After the transfer, PCR assays were performed on patients during the 1st, 7th, and 14th days. After 14 days of observation, patients with negative results on PCR and without other conditions in need of hospitalization were discharged. Re-positive cases were required to yield negative on consecutive PCR assays at an interval of at least 24 h.

### 2.1. Data Collection, Study Cohorts, and Outcome Measures 

Tianjin was the first city in mainland China to record Omicron variant infection. During the Omicron pandemic, all individuals, asymptomatic, with mild or typical Omicron symptoms, were tested and treated in isolation if found positive in a centralized facility. Treatment was critical to relieving symptoms and improving the disease outcomes.

The patient data collected included demographic information (age, gender, ICU admission and invasive mechanical ventilation (IMV) status, and admission/discharge date); the presence of underlying complications, including hypertension, DM, hyperlipidemia, coronary artery disease (CAD), congestive heart failure (CHF), chronic obstructive pulmonary disease (COPD), asthma, malignant tumor, liver disease, chronic kidney disease (CKD), and stroke; body mass index (BMI); COVID-19 vaccination history; symptoms (at admission and during hospitalization); disease severity; medication history and lifestyle; and PCR test results before admission and over the hospitalization period.

Height and weight were measured three times using standard anthropometric methods, and the average value was used in the subsequent analyses. Weight and height were measured while patients were wearing light indoor clothes. Blood pressure was measured in the supine or sitting position. BMI is calculated by dividing weight (kg) by the square of height (m^2^). Laboratory tests included fasting blood glucose level, triglyceride (TG) level, cholesterol level, routine blood test (leukocyte count, lymphocyte count, and neutrophil count), and the expression of inflammatory markers (CRP and IL-6), performed within 24 h of admission. All subjects underwent venous blood sampling after 12 h fasting. Serum routine blood test, liver and kidney function test, fasting blood glucose, lipid level, CRP, and IL-6 levels were measured with a full-automatic biochemical analyzer (7600A-020 Hitachi, Japan). The IgG and IgM antibodies against SARS-CoV-2 enveloping (E) protein and nucleocapsid (N) protein in serum samples were measured by chemiluminescence immunoassay. The IgM-IgG kits were purchased from Boosaic (Tianjin) Biotechnology Co., Ltd. (Tianjin, China).

This study’s participants were clustered into the MetS or non-MetS groups based on the modified WHO guidelines [15]. MetS was defined as having at least three of the following five factors: (1) prediabetes/DM (fasting blood glucose ≥ 5.6 mmol/L and/or glycated hemoglobin ≥ 5.7%) or history of DM; and (2) obesity (BMI ≥ 25 kg/m^2^). Since the proportion and distribution of body fat in Asian people differ from people in North America and Europe, the BMI calculation was performed based on Asian Obesity Standards (BMI ≥ 25 kg/m^2^) [16,17,18]. (3) Hypertension or use history of using anti-hypertension medicine. (4) TG ≥ 1.7 mmol/L, and (5) HDL < 1.0 mmol/L (female) and < 0.9 mmol/L (male) or use cholesterol-lowering medicine with a history of hypercholesterolemia. A patient was said to have DM and hypertension based on the history of the diseases, on treatment or usage of FBG ≥ 7.0 mmol/L, SBP ≥ 140, or DBP ≥ 90 mmHg. Dyslipidemia was considered under the following conditions: TC ≥ 6.22 mmol/L, TG ≥ 2.26 mmol/L, LDL-C ≥ 4.14 mmol/L, HDL-C ≤1.55 mmol/L, or use of anti-hyperlipidemic drugs [19].

The primary outcome of interest was the severity of COVID-19. According to the WHO COVID-19 clinical management life guideline, COVID-19 disease severity was classified into asymptomatic, mild, moderate, severe, and critical groups [20]. Asymptomatic infection and mild cases were classified as the ‘mild’ group, while moderate, severe, and critical cases were classified into the ‘severe’ group. The secondary outcome was intensive care unit (ICU) admission and PCR results in the recovery period. PCR re-positivity was defined as a positive PCR test with a Ct value <40 after two negative results in at least 24 h.

### 2.2. Laboratory Confirmation

SARS-CoV-2 PCR test was performed at two hospitals, Tianjin Haihe Hospital and Tianjin First Central Hospital, designated for isolating Omicron patients. The SARS-CoV-2 RNA was extracted from nasopharyngeal swabs using a commercial kit (Zybio, 5203050). Reverse transcription–polymerase chain reaction (RT-PCR) targeting the open reading frame of 1ab (ORF1ab) and nucleocapsid protein (N) [21] of the virus was performed following the WHO protocol. The circulation threshold (Ct value) less than 37 with an S-shaped amplification curve was considered a positive SARS-CoV-2 diagnosis, whereas Ct value ≥ 40 was considered a negative SARS-CoV-2 diagnosis. A retest was performed at 37 ≤ Ct < 40. The test was considered positive if ORF1ab and N were confirmed in the same sample using real-time RT-PCR. However, resampling and retesting were performed if only one of the two genes was detected. If only one of the two target genes was simultaneously detected in two samples. The patient was considered positive for SARS-CoV-2.

### 2.3. Statistical Analysis

Normally distributed continuous data were presented as the mean ± standard deviation, and skewed data as the median and interquartile range. Categorical variables were expressed as numbers and percentages (%). Student’s *t*-test was used for analyzing parametric continuous variables between the two groups. The Mann–Whitney U test was used to analyze non-parametric continuous variables. Categorical variables were analyzed using the χ^2^ test, with multiple comparisons across different groups. The MetS cohort was analyzed hierarchically after adjustment for age, gender, number of complications, and vaccination history. The relationship between the severity of COVID-19 and features of the MetS patients (DM, hypertension, obesity, and hyperlipidemia) and Omicron variant to calculate odds ratios (ORs) and 95% confidence intervals (CIs) was evaluated using logistic regression analysis. Data were analyzed using SPSS software, version 26 (IBM Corp, New York, NY, USA). Two-tailed *p* < 0.05 was considered statistically significant.

## 3. Results

### 3.1. Demographic and Clinical Characteristics of the Study Participants

A total of 316 adults with Omicron infection admitted in Tianjin at Tianjin Haihe Hospital and Tianjin First Central Hospital between 8 January and 7 February 2022 were recruited in this study. Of these, 111 (35.1%) patients were diagnosed with MetS. The average age of the study cohort was 46.7 ± 15.5 years, comprising 135 males (42.7%) and 181 females (57.3%). There was no difference in age (*p* = 0.123) and gender (*p* = 0.052) between the MetS and non-MetS groups, but the BMI was significantly higher in the MetS group (27.1 ± 4.0 Kg/m^2^ vs. 24.2 ± 3.9 Kg/m^2^ in the non-MetS group) (*p *< 0.01). Different complications were observed in 50.9% of the study participants and 25.2% and 13.2% of the MetS, and the non-MetS cohort presented with three or more complications. In the early hospitalization period, the serum concentrations of CRP, leucocyte, neutrophil, and monocyte were higher in the MetS group than in the non-MetS group, but there was no difference in the levels of SARS-CoV-2 antibodies (IgG/IgM) between the two groups. Monoclonal antibodies were used in only 5 patients in the sample. We observed no statistically significant differences between the two groups, as well as 31 patients for antiviral therapy. No difference was noted in CKD and cancer patients. Table 1 summarizes the demographic and clinical features of the study cohort.

Among the 316 study participants, 242 (76.6%) were administered with inactivated SARS-CoV-2 vaccine (BBIBP-CorV, CoronaVac, or other), whereas 50 (15.8%) were vaccinated with adenovirus vector vaccine (Ad5-nCoV), and 24 (7.6%) were unvaccinated. Most patients revealed mild (35.1%) and moderate (63.6%) symptoms, whereas 1% and 0.3% were severe and asymptomatic cases, respectively. None of the study participants died during the study period. Patient symptoms, including fever, fatigue, nasal congestion, cough, sore throat, runny nose, abnormal smell and taste, but not rash, diarrhea, and conjunctivitis, slightly worsened in the convalescence period (Table 2).

A small proportion of patients (5.4%) were admitted to the ICU during hospitalization, and the ICU admission was significantly higher in the MetS group (11.7% vs. 2.0%, *p* < 0.01). Only 0.9% of the whole cohort required ventilator-assisted mechanical ventilation, and no significant difference in this requirement was noted between the two groups. A patient was discharged from Haihe Hospital and transferred to the Tianjin First Central Hospital for rehabilitation treatment upon testing negative for SARS-CoV-2 RNA. During the rehabilitation period, patients received PCR tests on the 1st, 7th, and 14th days. Surprisingly, 71 (22.5%) of individuals in the rehabilitation treatment retested positive, and the proportion was significantly higher for the MetS group (32.4% vs. 17.1%, *p* < 0.01). 

### 3.2. Multivariate Analysis of Omicron Clinical Disease Outcome 

Age was a risk factor for poor disease outcomes and remained so even after adjustment for gender, number of complications, and vaccination status, binary logistic regression showed that age was associated with ICU admission (OR 1.11, 95% CI 1.04–1.19, *p* < 0.01), PCR re-positivity (OR 1.02, 95% CI 1.00–1.04, *p* = 0.03), and COVID-19 severity (OR 1.05, 95% CI 1.03–1.07, *p*  < 0.01). Gender and the number of underlying complications were not significantly linked to the above features. Table 3 summarizes the details of the relationship between Omicron clinical disease outcomes and patient factors.

Logistic regression analysis revealed a strong relationship between age and ICU admission (OR 1.08, 95% CI 1.00–1.17, *p* = 0.04), but not PCR re-positivity and COVID-19 severity. Table 4 summarizes the relationship between age, gender, number of underlying complications, and clinical outcomes (Admission to ICU, re-positive status, COVID-19 severity) of patients in the MetS group.

### 3.3. Correlation Analysis of Each Component in MetS with Severe COVID-19

In the adjusted analysis, MetS was strongly associated with ICU admission (OR 8.07, 95% CI 1.36–47.92, *p* = 0.02), PCR re-positivity (OR 3.06, 95% CI 1.65–5.68, *p* < 0.01), and severity of COVID-19 (OR 4.45, 95% CI 2.36–8.43, *p* < 0.01). For individual complications in the MetS group, prediabetes/DM was strongly associated with PCR re-positivity (OR 2.38, 95% CI 1.18–4.80, *p =* 0.02) and severity of COVID-19 (OR 2.60, 95% CI 1.15–5.88, *p* = 0.02), but not with ICU admission (OR 1.65, 95% CI 0.40–6.80, *p* = 0.49). Additionally, obesity (BMI ≥ 25) was associated with ICU admission (OR 18.14, 95% CI 1.70–193.91, *p* = 0.02) and severity of COVID-19 (OR 2.28, 95% CI 1.37–3.80, *p* < 0.01), but not PCR re-positivity (OR 0.80, 95% CI 0.46–1.39, *p* = 0.42). Hypertension and hyperlipidemia were linked to poor Omicron outcomes. Table 5 summarizes the relationship between patient parameters of the MetS group and the clinical outcomes of COVID-19.

### 3.4. The Efficacy of the Inactivated Vaccine against the Omicron Variant in the MetS Patients

A total of 78 individuals in the MetS group had been vaccinated with inactivated vaccine, whereas 12 were unvaccinated. Table 6 shows the relationship between the dose of inactivated vaccine and disease severity. ICU admission rates for unvaccinated individuals, the partially vaccinated cohort (two doses), and the fully vaccinated (three doses) cohort were 66.7%, 19.2%, and 0, respectively, which was statistically different (*p* < 0.01). 

Vaccination, full or partial, was also associated with significantly lower PCR re-positivity. The PCR re-positivity in the unvaccinated cases, partially vaccinated, and fully vaccinated cohorts were 66.7%, 38.5%, and 27.5%, respectively (*p* = 0.017). There was no statistical significance in clinical COVID-19 severity between the different doses of inactivated vaccine.

Vaccination history was also associated with the results of laboratory tests related to inflammation (Table 6). For instance, the CRP (3 doses vs. unvaccinated cases: <0.01) was significantly lower in the fully vaccinated (three doses) than in the unvaccinated group. Moreover, vaccinated individuals showed early recovery (early PCR negativity). However, no difference in IL-6 levels was noted between the two groups. The serum SARS-CoV-2 IgG level was highest among the partially vaccinated group (221.11 (191.81–247.27) S/CO). Moreover, the antibody titers for the partially and fully vaccinated group were significantly higher than those of the unvaccinated group. However, there was no significant difference in the IgM titters among the three groups.

## 4. Discussion

The present observational investigation of Omicron in patients was conducted in Tianjin, China, at the peak of the Omicron variant. In the context of mass vaccination, Omicron is less fatal than other SARS-CoV-2 variants. Multivariate analysis ICU admission rate, PCR re-positivity rate, and worse COVID-19 severity grade were higher in the MetS than in non-MetS patients after adjusting for age, gender, number of comorbidities, and vaccination history. This is consistent with some of the results of previous studies. 5,6,13 Subgroup multivariate analysis revealed that obesity and prediabetes/DM were independently linked to disease severity. These findings indicate that MetS is a compound high-risk factor associated with poor Omicron outcomes. Meanwhile, the predictive power of multiple MetS-related factors is higher than that of a single factor. Vaccination with inactivated vaccine, partial or full but especially full, is associated with a better prognosis of Omicron in MetS individuals.

Increasing evidence indicates that certain underlying diseases influence the clinical outcome of SARS-CoV-2 infection [22]. Obesity, hyperglycemia, hypertension, and dyslipidemia are the common features of MetS [23]. MetS increases the risk of cardiovascular disease by 1 fold and T2DM by 5 fold. Recent studies showed that severe COVID-19 is linked with impaired metabolism, and the risk of death is higher in the MetS cohort [24,25]. The prevalence of MetS in our Omicron cohort is 35.1%, consistent with the findings of a meta-analysis of more than 20,000 COVID-19 patients, suggesting that the prevalence of MetS is 3.6~47.1% (in COVID-19 patients) and 30.0~41.3% (in the general population in China) [26]. Herein, we found that Omicron patients in the MetS group had high leukocyte neutrophil and CRP, but there was no significant difference in IL-6 level and lymphocyte count between Omicron patients in the MetS and non-MetS groups. High levels of inflammatory mediators, such as IL-6, CRP, and neutrophil, have been reported in COVID-19 patients, and this reflects acute inflammatory responses related to cytokine storms. In patients with severe COVID-19, CD4 and CD8 levels were decreased. This suggested that lymphocytes can help to eliminate virally infected cells. Meanwhile, high lymphocyte count predicts better clinical outcomes of COVID-19 [6,27,28,29,30]. No study participant died in this work; nevertheless, a few individuals required assisted ventilation. The favorable Omicron outcome could be attributed to low inflammation induced by the Omicron variant and wide vaccination coverage in the city.

Previous research reported that the proportion of SARS-CoV-2 re-positivity after discharge was 2.4~69.2% [31]. Herein, the re-positivity ratio was 22.5%. Although SARS-CoV-2 re-positivity is a common phenomenon, the mechanisms of its occurrence remain unknown. IgM and IgG levels were comparable between the MetS and non-MetS groups, indicating that humoral immune response is not a dominant factor of SARS-CoV-2 re-positivity [32]. One previous study showed that a low level of CD8^+^ T cells is a predictor for delayed virus shedding [33]. Prolonged virus shedding and the lower IgG level are linked to SARS-CoV-2 re-positivity in COVID-19 patients [34], which, to some extent, explains the correlation between MetS and SARS-CoV-2 re-positivity.

The rehabilitation of Omicron patients involves the management of symptoms. Patient symptoms in the initial and recovery periods generally subside, however, symptoms involving multiple organs (e.g., diarrhea, rash, and conjunctivitis) do not completely disappear. Existing epidemiological data show that fever and respiratory symptoms precede gastrointestinal symptoms [35]. A study focusing on the gastrointestinal characteristics of COVID-19 revealed that after excluding possible drug-related factors, the incidence of diarrhea remained high at 22.2% [36]. The slightly higher incidence of rash and conjunctivitis in this work may be related to the prolonged inhalation of disinfectants in isolation facilities. However, additional studies are necessary to confirm this hypothesis. Meanwhile, in the convalescent period, the incidence of clinical symptoms was low. Because of the short follow-up, it is uncertain that the symptoms that had not subsided are “long-term” COVID-19-associated complications. Compared with previous SARS-CoV-2 variants, Omicron is associated with fewer “long-term” COVID-19-associated complications. Full vaccination significantly lowers the incidence of the above symptoms [37].

Binary logistic regression revealed that age is strongly associated with ICU admission in both the MetS and non-MetS groups. Specifically, the risk of a worse prognosis of Omicron variant infection increased with age even after vaccination [38]. In one meta-analysis of 611 study participants, the risk of death from COVID-19 increased with age, with the mortality rate of individuals below 50 years old being < 1%. However, this increased exponentially after 50 years. The highest mortality occurred among patients aged 80 years or older, which was six times higher than that in younger patients [39]. The high mortality rates for patients older than 80 years may be associated with physiological aging, low immunity, and other underlying diseases, although the specific underlying mechanism remains unclear. Older individuals are prone to other complications, decreasing their functional reserve and internal resilience. Additionally, several complications are extremely common among elderly patients, aggravating the severity of COVID-19 [40,41].

MetS increased the risk of ICU admission, PCR re-positivity, and worse Omicron variant-related COVID-19. Unlike a single high-risk factor, MetS is a better predictor of multiple adverse outcomes in Omicron variant-related COVID-19. Obesity (42%, 95% CI 34–49%), hypertension (40%, 95% CI 35–45%), and DM (17%, 95% CI 15–20%) are the most prevalent metabolism-related complications in patients with severe/fatal COVID-19 [42]. Correlation analysis confirmed that obesity increases the risk of ICU admission by 18.14 fold. Obesity is an independent predictor of other complications in patients with Omicron, and obese patients have a higher viral load, slower antiviral response, and worse disease progression [43,44]. Adipose tissue, a viral repository, is a highly active organ linking systemic immunity, endocrine, and metabolic homeostasis [45]. Obesity and prediabetes/DM increased the risk of worse disease progression by 2.28 and 2.6 times, respectively, consistent with the previous study. Specifically, ACE2 expression is upregulated in obese and DM patients, increasing the susceptibility to SARS-CoV-2 infection [46]. Obesity and DM-related pulmonary physiological abnormalities and microvascular diseases are associated with higher virus titer [47] and prolonged virus shedding [48], which might aggravate disease deterioration. On the other hand, SARS-CoV-2 infection may worsen underlying complications causing new endocrine and metabolic disorders, and the vicious circle continues. Hypertension also aggravates the severity and increases the risk of death from COVID-19 [49,50,51], in line with our findings. Studies on the relationship between dyslipidemia and COVID-19 prognosis are relatively few and inconsistent [52,53], potentially attributed to the complex pathology of dyslipidemia and the rapid change in the lipid profile [54]. Additionally, the pleiotropic effects of statin treatment (the main dyslipidemia treatment) include anti-inflammatory and antioxidant effects, which significantly reduce the severity and mortality due to COVID-19 [55]. These findings are particularly key because the high-risk factors for MetS are inconsistent. Generally, there is a need to manage early complications in high-risk groups, which may include regular pulmonary imaging and assessing the respiratory function, measuring biochemical indexes before the occurrence of adverse outcomes difficult to manage but significant to disease prognosis.

The interaction between COVID-19 and MetS is an important public health concern. Thus, there is a need to develop effective therapy and vaccines against these two diseases. The original COVID-19 vaccine is not extremely effective against Omicron and its mutant subtypes to varying degrees. Previous studies have shown that obesity and DM can damage immune memory (e.g., after flu vaccination), which may negatively impact the efficacy of the SARS-CoV-2 vaccine. Our study suggests that partial or full vaccination improves the disease outcome of Omicron and lowers the risk of ICU admission and PCR re-positivity. Vaccination also results in lower serum CRP and high IgG titer among MetS individuals. Evidence suggests that vaccination boosts induce additional plasma cell differentiation from the memory B cell compartment to improve neutralization efficacy [56] and T-cell responses are preserved across vaccine platforms, regardless of the variant [57]. Despite the lack of the Omicron variant-specific vaccine at this stage, the approved inactivated vaccine in China remains effective against this variant and can reduce related disease severity and mortality [58]. MetS increases the risk of Omicron infection. Identifying Omicron high-risk groups is particularly vital for targeted prevention/treatment, hence reducing the disease burden and related mortality.

This study has several limitations. First, considering its cross-sectional nature, causality analyses could not be performed. Meanwhile, whole genome sequencing (WGS) was only performed on the first two cases to verify the identity of the Omicron BA.1 pathogen. Other cases were believed to be Omicron BA.1 infection without further WGS, which might not have been the case. Additionally, due to the cross-sectional design, our sample size is relatively small, and extremely few individuals received only one vaccine shot, potentially causing false positive results, and influencing the validity of the outcomes. Therefore, additional studies are necessary to validate the findings. Despite these limitations, the current study provides valuable evidence to clarify that MetS is associated with poor Omicron infection prognosis while inactivated vaccine improves the outcome of Coronavirus disease 2019.

## 5. Conclusions

In conclusion, MetS increases the risk of ICU admission, PCR re-positivity, and severe COVID-19. It is also a composite predictor of poor outcomes of Omicron infection. Two shots of inactivated vaccine, particularly three doses, effectively protect against Omicron even among the high-risk group.

## Figures and Tables

**Figure 1 vaccines-11-01554-f001:**
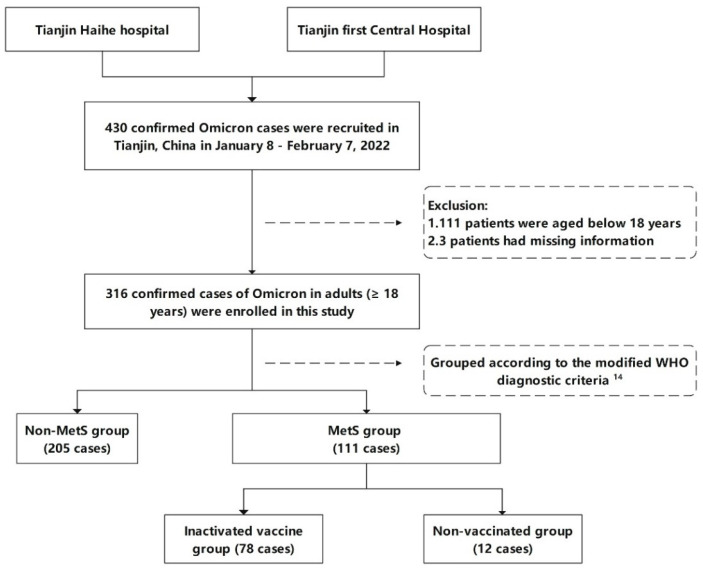
Flowchart.

**Table 1 vaccines-11-01554-t001:** Baseline characteristics of patients.

Characteristics	Total (*n* = 316)	MetS (*n* = 111)	Non-MetS (*n* = 205)	*p* Value
Age (years)	46.7 ± 15.5	48.7 ± 17.1	45.7 ± 14.5	0.123
Sex [*n* (%)]				0.052
Male	135(42.7)	52(50.5)	83(39.0)	
Female	181(57.3)	51(49.5)	130(61.0)	
BMI (Kg/m^2^)	25.2 ± 4.2	27.1 ± 4.0	24.2 ± 3.9	<0.01
IL-6 ≥ 1.5 (pg/mL)				0.293
Yes	102(32.3)	40(36.0)	62(30.2)	
No	214(67.7)	71(64.0)	143(69.8)	
CRP (mg/L)	0.96(0.40–1.91)	1.10(0.63–2.00)	0.79(0.28–1.85)	<0.01
Leukocyte (*10^9^/L)	6.20(5.24–7.38)	6.84(5.65–8.09)	6.03(5.02–6.89)	<0.01
monocyte (*10^9^/L)	0.42(0.35–0.51)	0.45(0.38–0.55)	0.41(0.34–0.50)	<0.01
lymphocyte (*10^9^/L)	2.04(1.61–2.43)	2.18(1.71–2.63)	1.96(1.60–2.34)	0.018
Neutrophil (*10^9^/L)	3.57(2.85–4.35)	3.79(3.09–4.85)	3.50(2.73–4.07)	<0.01
Comorbidities [*n* (%)]				0.01
None	155(49.1)	44(39.6)	111(54.1)	
1	64(20.3)	20(18.0)	44(21.5)	
2	42(13.3)	19(17.1)	23(11.2)	
≥3	55(17.4)	28(25.2)	27(13.2)	
COVID-19 severity ‡ [*n* (%)]				<0.01
Asymptomatic	1(0.3)	0(0.0)	1(0.5)	
Mild	111(35.1)	19(17.1)	92(44.9)	
Moderate	201(63.6)	89(80.2)	112(54.6)	
Severe	3(1.0)	3(2.7)	0(0.0)	
Critical	0(0.0)	0(0.0)	0(0.0)	
Admission of ICU [*n* (%)]				<0.01
Yes	17(5.4)	13(11.7)	4(2.0)	
No	299(94.6)	98(88.3)	201(98.0)	
Re-positive status [*n* (%)]				<0.01
Yes	71(22.5)	36(32.4)	35(17.1)	
No	245(77.5)	75(67.6)	170(82.9)	
Mechanical ventilation [*n* (%)]				0.588
Yes	3(0.9)	2(1.8)	1(0.5)	
No	313(99.1)	109(98.2)	204(99.5)	
Vaccination status [*n* (%)]				0.074
None	24(7.6)	12(10.8)	12(5.8)	
1 Dose of IV	2(0.6)	1(0.9)	1(0.5)	
2 Doses of IV	84(26.6)	26(23.4)	58(28.3)	
3 Doses of IV	156(49.4)	51(45.9)	105(51.2)	
1 Dose of AVV	15(4.7)	3(2.7)	13(6.3)	
2 Doses of AVV	35(11.1)	18(16.2)	16(7.8)	
IgG(S/CO)	206.7(178.1–235.8)	209.5(182.2–236.7)	204.6(173.7–234.5)	0.493
IgM(S/CO)	0.48(0.26–0.87)	0.48(0.26–0.77)	0.48(0.26–1.08)	0.560
Drugs				
Monoclonal antibodies [*n* (%)]	5(1.6)	2(1.8)	3(1.5)	0.73
antiviral [*n* (%)]	31(9.8)	13(11.7)	18(8.8)	0.104
CKD [*n* (%)]	94(29.7)	36(32.4)	58(28.2)	0.091
Cancer [*n* (%)]	6(1.9)	3(2.7)	3(1.5)	0.189

Note: COVID-19: coronavirus disease 2019; IV: inactivated vaccine; AVV: adenovirus-vectored vaccine. ‡ COVID-19 severity was defined according to WHO living guidance for clinical management of COVID-19.

**Table 2 vaccines-11-01554-t002:** Clinical symptoms at baseline and during the convalescent period *.

Symptoms	Initial	Convalescent
Fever	89(28.1)	11(3.5)
Fatigue	46(14.6)	21(6.6)
Nasal congestion	37(11.7)	17(5.4)
Cough	122(38.6)	24(7.6)
Sore throat	63(19.9)	15(4.7)
Rash	1(0.3)	3(0.9)
Nasal discharge	38(12.0)	21(6.6)
Diarrhea	4(1.3)	7(2.2)
Olfactory abnormalities	4(1.3)	2(0.6)
Taste abnormalities	5(1.6)	2(0.6)
Conjunctivitis	15(4.7)	21(6.6)

* The initial clinical symptoms were the ones experienced during hospitalization in Tianjin Haihe Hospital after a positive PCR result. During the rehabilitation period, clinical symptoms were the symptoms experienced after being discharged from Haihe Hospital and transferred to Tianjin First Central Hospital.

**Table 3 vaccines-11-01554-t003:** Multivariate analysis for the relationship between Omicron clinical disease outcomes and patient factors.

Characteristic	Admission of ICU			Re-Positive Status			COVID-19 Severity ‡		
	OR (95% CI)	Adjusted OR (95% CI)	*p*	OR (95% CI)	Adjusted OR (95% CI)	*p*	OR (95%CI)	Adjusted OR (95% CI)	*p*
**Age** **(years)**	1.14(1.09–1.20)	1.11(1.04–1.19)	<0.01	1.03(1.01–1.04)	1.02(1.00–1.04)	0.03	1.05(1.03–1.06)	1.05(1.03–1.07)	<0.01
**Gender**									
Female	Reference	Reference		Reference	Reference		Reference	Reference	
Male	0.54(0.19–1.58)	0.78(0.20–3.06)	0.72	0.67(0.39–1.15)	0.76(0.43–1.34)	0.34	0.84(0.53–1.33)	0.89(0.54–1.47)	0.63
**Comorbidities**									
Without	Reference	Reference		Reference	Reference		Reference	Reference	
With	16.99(2.23–129.77)	2.54(0.27–23.85)	0.41	1.06(0.63–1.80)	0.73(0.39–1.37)	0.33	1.66(1.04–2.64)	0.84(0.49–1.46)	0.53

Note: OR: odds ratio; CI: confidence interval; ICU admission, re-positive status, and COVID-19 severity adjusted for age, gender, comorbidities, and vaccination status. ‡ COVID-19 disease severity is classified into asymptomatic, mild, moderate, severe, and critical groups [20]. Asymptomatic infection and mild cases are classified as the ‘mild’ group, while moderate, severe, and critical cases are classified into the ‘severe’ group.

**Table 4 vaccines-11-01554-t004:** Association between MetS and COVID-19 severity and progression.

Characteristic	Admission of ICU			Re-Positive Status			COVID-19 Severity ‡		
	OR (95% CI)	Adjusted OR (95% CI)	*p*	OR (95% CI)	Adjusted OR (95% CI)	*p*	OR (95%CI)	Adjusted OR (95% CI)	*p*
**Age** **(years)**	1.13(1.06–1.21)	1.08(1.00– 1.17)	0.04	1.01(0.99–1.04)	1.01(0.97–1.04)	0.73	1.04(1.00–1.07)	1.03(0.99–1.07)	0.14
**Gender**									
Female	Reference	Reference		Reference	Reference		Reference	Reference	
Male	0.30(0.08–1.16)	1.47(0.17–12.43)	0.73	0.37(0.16–0.85)	0.46(0.18–1.17)	0.10	0.34(0.12–0.97)	0.38(0.12–1.21)	0.10
**Comorbidities**									
Without	Reference	Reference		Reference	Reference		Reference	Reference	
With	1.81(0.13–26.45)	1.78(0.17–19.26)	0.64	0.88(0.39–1.98)	0.81(0.27–2.47)	0.71	1.90(0.70–5.13)	1.31(0.38–4.51)	0.67

Note: Admission to ICU, re-positive status, and COVID-19 severity adjusted for age, gender, comorbidities, and vaccination status. ‡ COVID-19 disease severity is classified into asymptomatic, mild, moderate, severe, and critical groups [20]. Asymptomatic infection and mild cases are classified as the ‘mild’ group, while moderate, severe, and critical cases are classified into the ‘severe’ group.

**Table 5 vaccines-11-01554-t005:** The relationship between MetS patient factors and ICU admission, PCR re-positivity, and severity of COVID-19.

Characteristic	Admission of ICU			Re-Positive Status			COVID-19 Severity ‡		
	OR (95% CI)	Adjusted OR (95% CI)	*p*	OR (95% CI)	Adjusted OR (95% CI)	*p*	OR (95%CI)	Adjusted OR (95% CI)	*p*
**MetS**	6.67(2.12–20.98)	8.07(1.36–47.92)	0.02	2.33(1.36–4.00)	3.06(1.65–5.68)	<0.01	4.02(2.29–7.08)	4.45(2.36–8.43)	<0.01
Hypertension	2.79(0.89–8.74)	0.87(0.17–4.50)	0.87	0.86(0.51–1.46)	0.80(0.42–1.50)	0.48	1.30(0.82–2.06)	0.81(0.46–1.42)	0.46
Prediabetes/DM	6.34(2.31–17.39)	1.65(0.40–6.80)	0.49	2.54(1.40–4.61)	2.38(1.18–4.80)	0.02	4.33(2.05–9.14)	2.60(1.15–5.88)	0.02
Obesity	3.69(1.04–13.10)	18.14(1.70–193.91)	0.02	0.82(0.48–1.40)	0.80(0.46–1.39)	0.42	2.23(1.39–3.56)	2.28(1.37–3.80)	<0.01
Hyperlipidemia	0.79(0.30–2.11)	1.77(0.38–8.20)	0.46	0.68(0.40–1.16)	0.81(0.45–1.43)	0.46	1.05(0.66–1.66)	0.98(0.58–1.64)	0.93

Note: MetS= metabolic syndrome, adjusted for age, sex, comorbidities, and vaccination status; DM = diabetes mellitus, adjusted for age, sex, BMI, comorbidities, and vaccination status; obesity, (BMI ≥ 25), adjusted for age, sex, comorbidities, and vaccination status. ‡ COVID-19 disease severity is classified into asymptomatic, mild, moderate, severe, and critical groups [20]. Asymptomatic infection and mild cases are classified as the ‘mild’ group, whereas moderate, severe, and critical cases are classified into the ‘severe’ group.

**Table 6 vaccines-11-01554-t006:** The relationship between vaccination status and COVID-19 severity and progression.

	3 Doses of IV (*n* = 51)	2 Doses of IV (*n* = 26)	1 Dose of IV (*n* = 1)	Unvaccinated (*n* = 12)	*p*-Value	
Admission of ICU					<0.01	
Yes	0(0.0)	5(19.2)	0(0.0)	8(66.7)		
No	51(100.0)	21(80.8)	1(100.0)	4(33.3)		
Re-positive status					0.038	
Yes	14(27.5)	10(38.5)	1(100.0)	8(66.7)		
No	37(72.5)	16(61.5)	0(50.0)	4(33.3)		
COVID-19 severity ‡					0.051	
Asymptomatic	0(0.0)	0(0.0)	0(0.0)	0(0.0)		
Mild	0(0.0)	1(3.8)	0(0.0)	2(16.7)		
Moderate	39(76.5)	19(73.1)	1(100.0)	10(83.3)		
Severe	12(23.5)	6(23.1)	0(0.0)	0(0.0)		
Critical	0(0.0)	0(0.0)	0(0.0)	0(0.0)		
IL-6 ≥ 1.5 (pg/mL)					0.220	
Yes	17(33.3)	10(38.5)	1(100.0)	7(58.3)		
No	34(66.7)	16(61.5)	0(00.0)	5(41.7)		
CRP (mg/L)	1.02(0.58–1.43)	1.09(0.68–2.22)	-	3.29(0.91–41.92)	0.028	3 vs.Un: <0.01
Leukocyte (*10^9^/L)	6.94(5.63–7.91)	6.51(5.66–7.44)	-	7.63(4.85–8.53)	0.353	
Monocyte (*10^9^/L)	0.45(0.37–0.51)	0.46(0.38–0.63)	-	0.50(0.34–0.56)	0.553	
Lymphocyte (*10^9^/L)	2.19(1.71–2.73)	2.14(1.76–2.62)	-	1.67(0.80–2.75)	0.169	
Neutrophil (*10^9^/L)	3.89(3.15–4.75)	3.62(3.14–4.64)	-	3.62(2.91–4.66)	0.405	
IgG (S/CO)	209.41(182.21–235.24)	221.11(191.81–247.27)	-	168.45(2.26–184.07)	<0.01	3 vs.Un: <0.01 2 vs. Un: <0.01
IgM (S/CO)	0.50(0.30–0.83)	0.58(0.21–1.06)	-	0.36(0.19–0.53)	0.262	

Note: ‡ COVID-19 severity was defined according to WHO living guidance for clinical management of COVID-19.

## Data Availability

The datasets during the current study are available from the corresponding author upon reasonable request.

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
