# Peer review of "Metabolic Syndrome Is Associated with Poor Omicron Infection Prognosis While Inactivated Vaccine Improves the Outcome of Coronavirus Disease 2019 among Chinese Inhabitants: A Retrospective Observational Study from a Chinese Municipality"

_vaccines, 2023, doi:10.3390/vaccines11101554_

Round 1
Reviewer 1 Report
The work entitled “Metabolic syndrome is associated with poor Omicron
infection prognosis while inactivated vaccine improves the outcome of
Coronavirus disease 2019: A retrospective observational study." examines
retrospectively the relationship between metabolic syndrome and poor
COVID- Omicron strain infection prognosis, but also provides insights into
the plausible disadvantage (of people with metabolic syndrome) in achieving
protective immunity after vaccination. The manuscript brought interesting
comparisons between available data at Chinese Isolation public hospitals in
Tianjin. The paper is well written, clearly and understandable also by a non
professional audience. Nevertheless, some acronyms should be clarified for
everybody that is not familiar with the medical jargon, for example ICU
(allegedly Intensive Unit Care, -please make it clear in the text-). Also all
abbreviations appearing in the abstract should be clarified in the abstract
beforehand.
Some other corrections:
Correct inpatiens to in patiens in line 251
Correct “inconsistent” to “in a consistent manner” in line 335.
Correct “pleiotropy” to “pleiotropic effects” in line 338.
Define WGS in line 363.
Once these corrections are made, the paper is ready for publication.
Overall completely understandable. Some minor mistakes/typos to be corrected.
Author Response
Dear reviewers,
Thank you very much for your comments and professional advice. These opinions help to improve academic rigor of our article. Based on your suggestion and request, we have made corrected modifications on the revised manuscript. Meanwhile, the manuscript had be reviewed and edited by language services of MJEditor (www.mjeditor.com). We hope that our work can be improved again. Furthermore, we would like to show the details as follows:
Reviewer 1#
1.The paper is well written, clearly and understandable also by a non professional audience. Nevertheless, some acronyms should be clarified for everybody that is not familiar with the medical jargon, for example ICU (allegedly Intensive Unit Care, -please make it clear in the text-). Also all abbreviations appearing in the abstract should be clarified in the abstract beforehand. Some other corrections:
Correct inpatients to in patients in line 251
Correct “inconsistent” to “in a consistent manner” in line 335.
Correct “pleiotropy” to “pleiotropic effects” in line 338.
Define WGS in line 363.
Once these corrections are made, the paper is ready for publication.
The author’s answer: We thank the reviewer for pointing these details out. We agree and have revised as follows(marked with a yellow background):
1.1 In the abstract, we clarified all abbreviations appearing, for example COVID-19 and MetS in the line 12.ICU and PCR in the line 21.ACE2 in the line 34.
1.2 Correct inpatients to in patients in line 283
1.3 Correct “inconsistent” to “in a consistent manner” in line 369.
1.4 Correct “pleiotropy” to “pleiotropic effects” in line 372.
1.5 Define WGS in line 397.
We would like to thank the referee again for taking the time to review our manuscript. Look forward to hearing from you.
Reviewer 2 Report
1) In Introduction, the authors should provide information on the relationship of COVID-19, MetS, and Vaccination on the outcome of COVID-19 infection. Besides, the authors should mention, why they have considered this study.
2) The authors presented omicron infection throughout the manuscript as the COVID-19. This is confusing. As Omicron is a variant of SARS-CoV-2, it is better to present the Omicron case as Omicron-COVID-19 or Omicron infection. Please correct the whole manuscript thoroughly. This is very essential to search for any difference on the metS, ICU and prevalence of the reduction of metS by SARS-CoV-2 and Omicron.
3) In Methodology, the authors should confirm the test of COVID-19 by Omicron, instead of general description of the confirmation of COVID-19.
4) The authors should discuss on the outcomes of their present study with that of the previous studies with SARS-CoV-2 by other researchers as well as similar other studies conducted by other researchers which is lacking in this manuscript.
5) The number of Omicron cases in this retrospective study is also very less. As this is a retrospective study, the number of patients should be increased.
The authors should thoroughly revise the manuscript for English grammar and language improvement.
Author Response
Dear reviewers,
Thank you very much for your comments and professional advice. These opinions help to improve academic rigor of our article. Based on your suggestion and request, we have made corrected modifications on the revised manuscript. Meanwhile, the manuscript had be reviewed and edited by language services of MJEditor (www.mjeditor.com). We hope that our work can be improved again. Furthermore, we would like to show the details as follows:
Comments and Suggestions for Authors
1.In Introduction, the authors should provide information on the relationship of COVID-19, MetS, and Vaccination on the outcome of COVID-19 infection. Besides, the authors should mention, why they have considered this study.
The author’s answer: We agree with the reviewer that further elaborating on this point would be helpful. We mentioned “the relationship of COVID-19, MetS, and Vaccination on the outcome of COVID-19 infection” in the original article as follows: [Increasing evidence has linked MetS to severe coronavirus disease 2019 (COVID-19). Accordingly, MetS has been proposed as a predictor of poor outcomes of COVID-19.]. However, in order to better demonstrate the purpose of our study, we add the following sentence to paragraph 2 and paragraph 3 in Introduction: 1)All vaccines appear to be safe and effective tools to prevent severe COVID-19, hospitalization, and death against all variants of concern, but the quality of evidence greatly varies. 2) We found inconsistent prognosis, particularly in patients with metabolic disorders.
2.The authors presented omicron infection throughout the manuscript as the COVID-19. This is confusing. As Omicron is a variant of SARS-CoV-2, it is better to present the Omicron case as Omicron-COVID-19 or Omicron infection. Please correct the whole manuscript thoroughly. This is very essential to search for any difference on the metS, ICU and prevalence of the reduction of metS by SARS-CoV-2 and Omicron.
The author’s answer: This observation is correct. We have changed necessary COVID-19 to Omicron-COVID-19, a total of 24 places (marked with a blue background) in the whole manuscript.
- In Methodology, the authors should confirm the test of COVID-19 by Omicron, instead of general description of the confirmation of COVID-19.
The author’s answer: Thanks to the author for the detailed review of the article, Because China is in a period of strict quarantine during the collection of cases, whether it is between cities or within the hospital, there is no large-scale movement of people and the diversity of virus variations, so whole genome sequencing (WGS) was performed on the first two cases to confirm the identity of the Omicron BA.1 pathogen. Other cases were believed to be Omicron BA.1 infection. Of course, we do not deny that it may affect the final result, and we have written it into the limitations of the article.
4.The authors should discuss on the outcomes of their present study with that of the previous studies with SARS-CoV-2 by other researchers as well as similar other studies conducted by other researchers which is lacking in this manuscript.
The author’s answer: We agree and have updated. In Discussion, we compare the conclusions of the article with a number of previous studies in detail, and add the following sentence [consistent with some of the results of previous studies.]to paragraph 1 which based on references 5,6,13.
5.The number of Omicron cases in this retrospective study is also very less. As this is a retrospective study, the number of patients should be increased.
The author’s answer: We agree with the reviewer that further elaborating on this point using new data would be helpful. However, we have completely collected all cases of Omicron infection, a total of 318 adult cases infected with Omicron were recorded between January 8 and February 7, 2022, with no new infections detected for the following 16 days during mass PCR testing. and although there are relevant datas for Omicron in other periods, the WGS of the virus may not be completely consistent. In order to better ensure the accuracy of our data, we chose not to make this change. And we look up an article, named Metabolic Syndrome and COVID-19 Mortality Among Adult Black Patients in New Orleans, which published Diabetes Care, Data were collected from 287 consecutive patients with COVID-19. But we remain added the limitations to paragraph 9 in the discussion: [Additionally, in this study, due to the cross-sectional design, our sample size is relatively small and very few individuals received only one vaccine shot, which may cause false positive results, and affect the validity of our results. further studies are needed to confirm. Despite these limitations, the current study provides valuable evidence to clarify MetS is associated with poor Omicron infection prognosis while inactivated vaccine improves the outcome of Coronavirus disease 2019.]
6.Comments on the Quality of English Language
The authors should thoroughly revise the manuscript for English grammar and language improvement.
The author’s answer: We appreciate the reviewer’s intimate advice, it will promote the article to be more perfect. The manuscript had be reviewed and edited by language services of MJEditor (www.mjeditor.com). We hope our hard work can be improved our article better. If reviewer feel that our article needs to be adjusted English grammar and language again, we are willing to revise it in anytime.
We would like to thank the referee again for taking the time to review our manuscript. Look forward to hearing from you.
Reviewer 3 Report
I was invited to revise the paper entitled "Metabolic syndrome is associated with poor Omicron infection prognosis while inactivated vaccine improves the outcome of Coronavirus disease 2019: A retrospective observational study". It was as cross-sectional study aimed to evaluate the association between metabolic syndrome and covid-19 outcomes.
Despite the paper is interesting, the present paper presents several limitations:
- Introduction section was poor and should report also information on vaccinaztion campaign occurred in the Tianjin Municipality;
- The title can led to misunderstanding: poor outcomes were associated to low efficacy vaccines. So Authors have to modify it highlightening that the present study was performed in CHina. I suggest " "Metabolic syndrome is associated with poor Omicron infection prognosis while inactivated vaccine improves the outcome of Coronavirus disease 2019 among Chinese in-habitants: A retrospective observational study from a Chinese Municipality;
- Enrollment procedure should be better described in methods section;
- Statistical analysis section nnedsz improvements. Mann-Whitney U test was used for non normally distributed data. In addition, correction for multiple comparisons should be performed;
- It is unclea which is the dependent variable of Table 4;
- It is unclear why IL-6 was dicotomized. Other laboratory variables were presented as continous;
- Drug therapy performed by each patient can influence the study outcome (monoclonal antibodies, antiviral etc) so, without this information, results are flawed!!
- Despite Authors divided patients by METS, Authors should also report the presence of each comorbidity: diabetes, hypertension, obesity etcs both in table 1 and in multivariate models. These variables are known risk factors for covid lethality and worse outcomes (10.3390/v15091794);
- Authors did not considered other important covariates as risk factors such as CKD and cancer;
- Authors should also evaluate the re-infection rate. Patients with previous infection could impact study outcomes.
Finally, this study was performed in a setting with a low coverage of effective vaccination.
Author Response
Dear reviewers,
Thank you very much for your comments and professional advice. These opinions help to improve academic rigor of our article. Based on your suggestion and request, we have made corrected modifications on the revised manuscript. Meanwhile, the manuscript had be reviewed and edited by language services of MJEditor (www.mjeditor.com). We hope that our work can be improved again. Furthermore, we would like to show the details as follows:
Despite the paper is interesting, the present paper presents several limitations:
1.Introduction section was poor and should report also information on vaccinaztion campaign occurred in the Tianjin Municipality;
The author’s answer: We have made the change. The new sentence in the line 54 reads as follows(marked with a purple background): Mass vaccination programs have been rolled out globally over the past 2 years. In the Chinese mainland, the most delivered vaccines are inactivated vaccines. As of January 8, 2022, when Omicron first emerged in Tianjin, up to 93.2% of its residents had been vaccinated to a varied extent. Citing a previous article published in Cell Research, please see the References [14]: Zheng, H., et al., Disease profile and plasma neutralizing activity of post-vaccination Omicron BA.1 infection in Tianjin, China: a retrospective study. Cell Res, 2022. 32(8): p. 781-784.
2.The title can led to misunderstanding: poor outcomes were associated to low efficacy vaccines. So Authors have to modify it highlightening that the present study was performed in CHina. I suggest " "Metabolic syndrome is associated with poor Omicron infection prognosis while inactivated vaccine improves the outcome of Coronavirus disease 2019 among Chinese in-habitants: A retrospective observational study from a Chinese Municipality;
The author’s answer: This observation is correct. We have changed the title as reviewer’s suggestion. This revision makes the title of the article more targeted.
3.Enrollment procedure should be better described in methods section
The author’s answer: We appreciate the reviewer’s insightful suggestion and agree that it would be useful to demonstrate, so we added the following paragraphs(marked with a purple background) in the line 77:
[All patients, including asymptomatic and mild cases, were hospitalized in Tianjin Haihe Hospital upon positive PCR results. Patients were discharged from Tianjin Haihe Hospital if the following criteria were met: 1) body temperature restored and stayed normal for over 3 days; 2) respiratory symptoms significantly relieved; 3) acute exudation substantially resolved on imaging study of the lungs; 4) negative on two consecutive PCR tests (at an interval of at least 24 hours) of samples collected from the respiratory tract. For patients whose PCR assays remained positive for over 4 weeks after criteria 1), 2) and 3) had been met, antibody assay and virus culture were applied to assess the risk of transmission before deciding whether these patients could be discharged.
Discharged patients from Tianjin Haihe Hospital were then transferred to Tianjin First Central Hospital for at least 14 days under medical observation. Patients received PCR assays on the 1st, 7th and 14th days after being transferred to Tianjin First Central Hospital. After 14 days of observation, patients with negative results on PCR and without other conditions in need of hospitalization were discharged. Re-positive cases were required to yield negative on consecutive PCR assays at an interval of at least 24 hours.]
4.Statistical analysis section need improvements. Mann-Whitney U test was used for non normally distributed data. In addition, correction for multiple comparisons should be performed;
The author’s answer: We apologize if our original statistical description make reviewer’s confused. We have modified the expression and hope that it is now clear. The new sentence reads as follows(from line 157 to line 162):
[Normally distributed continuous data are expressed as the mean ± standard deviation, and skewed data as the median and interquartile range. Categorical variables are expressed as numbers and percentages (%). Student's t-test was used for analysing parametric continuous variables between the two groups. The Mann–Whitney U test was used for analysing non-parametric continuous variables. Categorical variables were analysed using the χ2 test,with multiple comparisons across different groups.]
5.It is unclear which is the dependent variable of Table 4;
The author’s answer: We have fixed the dependent variable in the description and hope that it is now clearer.. Please see page 7 of the revised manuscript, lines 232–233.
6.It is unclear why IL-6 was dicotomized. Other laboratory variables were presented as continous;
The author’s answer: Thanks to the reviewers for asking this question, which can be confusing. The measured value of IL-6 in our laboratory is a non-continuous variable, and less than 1.5 cannot accurately measure the specific value, so we treat it as a categorical change in statistics. This is consistent with previously published article of our hospital. Please see the References [14]: Zheng, H., et al., Disease profile and plasma neutralizing activity of post-vaccination Omicron BA.1 infection in Tianjin, China: a retrospective study. Cell Res, 2022. 32(8): p. 781-784.
7.Drug therapy performed by each patient can influence the study outcome (monoclonal antibodies, antiviral etc) so, without this information, results are flawed!!
The author’s answer: We have revised the datas to address your concerns. Please see the Table 1 in Page 6. After statistical analysis, we see no differences between two groups patients treated with monoclonal antibodies and antiviral therapy. And We modified the statistical description in Page 5 line 183 to 186.
8.Despite Authors divided patients by METS, Authors should also report the presence of each comorbidity: diabetes, hypertension, obesity etc. both in table 1 and in multivariate models. These variables are known risk factors for covid lethality and worse outcomes (10.3390 v15.091794);
The author’s answer: We apologize if our Table 1 did not show the presence of each comorbidity: diabetes, hypertension, obesity etc. however, We believe that the proportion of comorbidities in Table 1 and the subsequent data analysis be discussed in article can meet the research purpose of the article without supplementation. We hope this was the right understanding.
9.Authors did not considered other important covariates as risk factors such as CKD and cancer;
The author’s answer: According to the reviewers' opinions, We have added the datas in the Table1,including CKD and cancer patients. Although some datas were added, none of them had an differentiated impact on the results between groups.
10.Authors should also evaluate the re-infection rate. Patients with previous infection could impact study outcomes. Finally, this study was performed in a setting with a low coverage of effective vaccination.
The author’s answer: We appreciate the reviewer’s insightful suggestion, however, there is no agreement on the threshold of Ct value for positive results. For example, a Ct value ≥ 35 would be regarded as negative in the US, Canada and Japan, and ≥ 30 in Germany.(References: Abu-Raddad LJ, Chemaitelly H, Ayoub HH, et al. Relative infectiousness of SARSCoV-2 vaccine breakthrough infections, reinfections, and primary infections. Nature communications 2022; 13(1): 532.) Even when re-positive patients are defined by such standards, No secondary infection was caused by the re-positive cases and no recurrence or worsening of symptoms was observed, which is a major difference from reinfection cases. This indicates that discrete elevation in Ct value is a cross-sectional manifestation over the course of viral infection. Positive results on RT-PCR assay indicate the presence of nucleic acid fragments of the virus, but it does not necessarily suggest the presence of whole viruses or replication ability of the virus. Such an analysis is beyond the scope of our paper, which aims only to show that MetS increases the risk of ICU admission, PCR re-positivity and severe COVID-19.
We would like to thank the referee again for taking the time to review our manuscript. Look forward to hearing from you.
Round 2
Reviewer 3 Report
It can be accepted for publication
Author Response
Dear reviewers,
Thank you very much for your comments and professional advice. These opinions help to improve academic rigor of our article. Based on your suggestion and request, we have made corrected modifications on the revised manuscript. Meanwhile, the manuscript had be reviewed and edited by language services of MJEditor (www.mjeditor.com). We hope that our work can be improved again.
We would like to thank the referee again for taking the time to review our manuscript. Look forward to hearing from you.
Yours sincerely,
Ying Liu